# ADAM17 Inhibition Increases the Impact of Cisplatin Treatment in Ovarian Cancer Spheroids

**DOI:** 10.3390/cancers13092039

**Published:** 2021-04-23

**Authors:** Nina Hedemann, Andreas Herz, Jan Hendrik Schiepanski, Jan Dittrich, Susanne Sebens, Astrid Dempfle, Julia Feuerborn, Christoph Rogmans, Nils Tribian, Inken Flörkemeier, Jörg Weimer, Sandra Krüger, Nicolai Maass, Dirk O. Bauerschlag

**Affiliations:** 1Department of Gynecology and Obstetrics, Kiel University and University Medical Center Schleswig-Holstein Campus Kiel, 24105 Kiel, Germany; stu127893@mail.uni-kiel.de (A.H.); stu204830@mail.uni-kiel.de (J.H.S.); stu211375@mail.uni-kiel.de (J.D.); Julia.Feuerborn@uksh.de (J.F.); Christoph.Rogmans@uksh.de (C.R.); stu122057@mail.uni-kiel.de (N.T.); Inken.Floerkemeier@uksh.de (I.F.); Joerg-Paul.Weimer@uksh.de (J.W.); Nicolai.Maass@uksh.de (N.M.); Dirk.Bauerschlag@uksh.de (D.O.B.); 2Institute for Experimental Cancer Research, Kiel University and University Medical Center Schleswig-Holstein Campus Kiel, 24105 Kiel, Germany; susanne.sebens@email.uni-kiel.de; 3Institute of Medical Informatics and Statistics, Kiel University and University Medical Center Schleswig-Holstein Campus Kiel, 24105 Kiel, Germany; dempfle@medinfo.uni-kiel.de; 4Department of Pathology, Kiel University and University Medical Center, Schleswig-Holstein Campus Kiel, 24105 Kiel, Germany; Sandra.Krueger@uksh.de

**Keywords:** ovarian cancer, chemotherapy resistance, combined therapy, ADAM17, 3D model, drug testing, spheroids, primary cells, primary spheroids, multicontent readout

## Abstract

**Simple Summary:**

Ovarian cancer (OvCa) treatment is still a challenge, mainly due to acquired resistance mechanisms during the course of chemotherapy. Here, we show the enhanced cytotoxicity of the combined treatment with the ADAM17 inhibitor GW280264X and cisplatin in comparison with cisplatin monotherapy. This effect was visible in five of five ovarian cancer cell lines grown as a monolayer and two of three tested cell lines in three-dimensional tumor spheroids. Tumor spheroids derived from primary tumor and ascites cells were sensitized to cisplatin treatment by GW280264X. In summary, the combination of ADAM17 inhibition with conventional chemotherapy seems to be a promising strategy to overcome chemotherapy resistance in OvCa.

**Abstract:**

Chemotherapy resistance is a major challenge in ovarian cancer (OvCa). Thus, novel treatment combinations are highly warranted. However, many promising drug candidates tested in two-dimensional (2D) cell culture have not proved successful in the clinic. For this reason, we analyzed our drug combination not only in monolayers but also in three-dimensional (3D) tumor spheroids. One potential therapeutic target for OvCa is A disintegrin and metalloprotease 17 (ADAM17). ADAM17 can be activated by chemotherapeutics, which leads to enhanced tumor growth due to concomitant substrate cleavage. Therefore, blocking ADAM17 during chemotherapy may overcome resistance. Here, we tested the effect of the ADAM17 inhibitor GW280264X in combination with cisplatin on ovarian cancer cells in 2D and 3D. In 2D, the effect on five cell lines was analyzed with two readouts. Three of these cell lines formed dense aggregates or spheroids (HEY, SKOV-3, and OVCAR-8) in 3D and the treatment effect was analyzed with a multicontent readout (cytotoxicity, viability, and caspase3/7 activation). We tested the combined therapy on tumor spheroids derived from primary patient cells. In 2D, we found a significant reduction in the half minimal (50%) inhibitory concentration (IC_50_) value of the combined treatment (GW280264X plus cisplatin) in comparison with cisplatin monotherapy in all five cell lines with both 2D readout assays (viability and caspase activation). In contrast, the combined treatment only showed an IC_50_ reduction in HEY and OVCAR-8 3D tumor spheroid models using caspase3/7 activity or Celltox^TM^ Green as the readout. Finally, we found an improved effect of GW280264X with cisplatin in tumor spheroids derived from patient samples. In summary, we demonstrate that ADAM17 inhibition is a promising treatment strategy in ovarian cancer.

## 1. Introduction

Ovarian cancer (OvCa) is one of the most common gynecological malignancies [1]. In the U.S., the majority (76%) of women are diagnosed in an advanced stage Fédération Internationale de Gynécologie et d’Obstétrique III and IV (FIGO III and IV) when disease has already spread within the abdominal cavity. Most patients develop chemotherapy resistance during the course of therapy, which is a major drawback of current chemotherapeutics [2]. In addition, the average relative 5-year survival rate of 43% in the U.S. indicates a need for more effective strategies to improve patient outcomes [3].

Over the last few decades, only a few breakthroughs have occurred regarding the treatment of OvCa. One of them was the introduction of poly-ADP-ribose-polymerase (PARP) inhibition in patients with a mutation in breast cancer gene 1/2 (BRCA1/2) and homologous recombination deficiency (HRD) [4]. Additionally, the antiangiogenic monoclonal antibody bevacizumab improves progression-free survival in advanced stages of OvCa [5]. Nevertheless, resistance to chemotherapy is still the major challenge in OvCa [1]. Several studies have dealt with potential resistance mechanisms, including enhanced platinum export via efflux pumps, differential expression of pro- or antitumoral proteins, and enhanced activation of growth factor receptors, such as receptor tyrosine kinases (RTKs) [6,7,8]. We previously showed that, in addition to well-known RTK activation by amplifications or mutations [9,10], a major activator of these growth factor receptors is A disintegrin and metalloprotease 17 (ADAM17). This protease is stimulated by chemotherapeutics such as cisplatin and might thus be an important novel target [11].

ADAM17 is physiologically involved in a variety of important signaling pathways inducing regeneration and organ development. However, in pathophysiological conditions, the over activation of this protease leads to enhanced substrate shedding and receptor activation. These substrates include, amongst others, a multitude of RTK-ligands, such as amphiregulin (AREG) and heparin-binding EGF-like growth factor (HB-EGF), leading to enhanced receptor activation [12]. In addition to well-known ADAM17 stimuli such as adenosine triphosphate (ATP) or lipopolysaccharide (LPS), ADAM17 can be activated by apoptosis [13,14]. Therefore, we previously investigated the effect of apoptosis on ADAM17 activity in OvCa cells [11]. Strikingly, we found a massive induction of ligand shedding once cells responded to cisplatin. This effect was reversed by ADAM17 inhibition using the inhibitor GW280264X, which specifically inhibits the metalloproteases ADAM10 and ADAM17, whereas the sole inhibition of ADAM10 by GI254023X did not inhibit the substrate release. This finding seems to be of major importance, as cisplatin-induced activation of ADAM17 leads to cellular survival responses, thus being a potential source of resistance development. Based on these observations, ADAM17 or its downstream signaling might be a valuable target for combinatorial therapies [15].

Even though some resistance pathways have been well-studied, the translation of in vitro studies into in vivo strategies often fails due to the lack of reliable test systems. Only 10% of drugs tested in preclinical models are later applied to patients [16]. As this may in part be due to the cellular system in which these substances were tested, we aimed to transfer the results of our monolayer studies to a more complex three-dimensional (3D) system. Two-dimensional (2D) monolayer cell culture models are widely used for the investigation of anticancer therapeutics. These systems are well-known for their beneficial characteristics, such as low-cost feasibility, high-throughput, ease of implementation, and regulation of microenvironmental factors, and are thus a good starting point to test cell behavior and treatment responses. Nevertheless, these models do not adequately reflect the in vivo situation of tumors [17,18,19]. Tumors are composed of several cell layers with intercellular contacts, and different growth properties and nutrition zones (i.e., hypoxic, proliferating, and senescent). Importantly, diffusion properties, polarization, and target expression might be affected, thus biasing the exclusive drug testing in 2D cell cultures. Hence, the reactions of cells in real tumors toward therapeutics might substantially differ.

Therefore, more realistic cell culture models that better mimic tumor physiology are warranted for testing innovative anticancer therapeutics and treatment combinations. One promising model for substance testing is the 3D tumor spheroid model. This 3D system has been used successfully for oncologic studies including the imitation of chemoresistance [17,20]. Within tumor spheroids, cancer cells display a 3D architecture, showing gradients of oxygen, nutrients, and pH analogous with key features of tumor biology in solid tumors [21]. Different zones, including a necrotic core, a senescent zone, and a peripheral proliferation zone are often described [22]. Some comprehensive studies have characterized the growth of OvCa cell lines in 2D and 3D [23,24]. Moreover, several publications focused on the treatment effects on tumor spheroids using single drug applications [25,26]. Hence, we developed a defined readout sequence in particular focusing on combinatorial treatments, being adaptable to established and primary OvCa cells.

The aim of this study was to analyze whether ADAM17 inhibition also sensitizes cells to cisplatin in a more complex 3D tumor spheroid model. After establishing the model, we combined different live–dead assays (CellTox™ Green Cytotoxicity Assay, Caspase-Glo^®^ 3/7 Assay System, and RealTime-Glo™ MT Cell Viability Assay) in one plate allowing for a high-content readout. Interestingly, small tumor spheroids can be found in ascites of OvCa patients capable of leading to intra-abdominal metastasis [27], and are thus of particular interest in this tumor entity. Apart from using cisplatin-sensitive and -resistant OvCa cell lines, we adapted our system to primary cells derived from tumor tissue and ascites of late-stage OvCa patient specimens. We provide strong evidence that the inhibition of ADAM17 is crucial for sensitizing cells to cisplatin, thus being an interesting target to be considered for combinatorial treatments.

## 2. Materials and Methods

### 2.1. Ethics Statement

According to the Declaration of Helsinki, this research was approved by the Institutional Review Board of the University Medical Center Schleswig–Holstein, Campus Kiel (AZ: B327/10). Written informed consent was obtained from all patients.

### 2.2. Cell Culture and Isolation of Primary Cells

OvCa cell lines HEY, Igrov-1, SKOV-3, and OVCAR-8, purchased from American Type Culture Collection (ATCC) and A2780 from Sigma-Aldrich (St. Louis, MO, USA), were cultured in Roswell Park Memorial Institute (RPMI)-1640 medium including L-glutamine (Sigma-Aldrich, #R8758) with 10% fetal bovine serum (Biochrom, Cambridge, UK) and penicillin–streptomycin (pen.-str.) (3000 U pen./30,000 μg str. per 500 mL RPMI-1640; Biochrom) at 37 °C and 5% CO_2_ in a humidified incubator, and subcultivated when a confluency of 70–80% was reached.

Primary tumor cells were extracted from tumor tissue of advanced-stage ovarian cancer patients during surgery as described previously [28]. To isolate primary ascites cells, ascites fluid was centrifuged (348× *g*, 10 min). The remaining pellet was resolved in 12 mL RPMI-1640 medium, supplemented as described above. Tumor- and ascites-derived cells were cultured in tissue culture flasks, expanded, and used for experiments between passage P1 and P4 once a confluency of ~75% was reached.

Mycoplasma contamination was routinely investigated using MycoAlert™ (Lonza, #LT07), and all cell lines and primary cells were authenticated by short tandem repeat (STR) DNA profiling analysis, as previously described [29].

### 2.3. Generation and Classification of Tumor Spheroids

Cells were harvested at 70–80% confluence using trypsin and counted using a Neubauer Chamber. OvCa cells or primary cells per well were seeded in ultra-low attachment (ULA) plates (Corning #7007) in the following cell numbers per well: A2780: 300; Igrov-1: 450; HEY: 300; SKOV-3: 6500; OVCAR-8: 4000; UF-354 ASC: 3000; and UF-354 TU: 1500. To do so, 150 µL medium was suspended in all wells, and 50 µL of pre-diluted cell suspension was added. Spheroid formation was imaged daily using NYONE^®^ Scientific (SYNENTEC) using 4× and 10× magnification. Plate-movement XY was set to “very gentle” to prevent spheroid movement during measurement. To classify cell aggregates and spheroids, cells were stained on day six. For the staining, half of the medium was removed and a staining solution with 2× final concentration of the following dyes in the medium were added. Calcein-AM (BioLegend, #425201), final concentration of 0.1 µg/mL, was used to stain living cells; propidium iodide (PI) (BioLegend, #421301), final concentration of 5 µg/mL, used to stain dead cells; and Hoechst33342 (Invitrogen, #H1399), final concentration of 1 µg/mL, used as a nuclear counterstain.

After 3 h incubation at room temperature (RT), the plate was imaged using the 10× objective of NYONE^®^ Scientific (SYNENTEC) and the settings: brightfield: Ex: BF; Em: Green (530/43 nm); Hoechst33342: Ex: UV (377/50 nm); Em: Blue (452/45 nm); Calcein-AM: Ex: Blue (475/28 nm); Em: Green (530/43 nm); PI: Ex: Lime (562/40 nm); Em: Red (628/32 nm). Three different focal offsets were used for each channel as the best focal plane might vary between the spheroids. Representative images of one focal plane are shown. A scale bar was added using ImageJ (Wayne Rasband (NIH)) and pictures were scaled using GIMP 2.10.14 (GNU Image Manipulation Program).

### 2.4. Drug Application and Subsequent Quantification of Treatment Effects and Substrate Release

#### 2.4.1. Drug Treatment and Analysis of Viability and Caspase Activity in 2D Cell Culture

Cells were harvested and counted as described above. We seeded 5000–10,000 cells per well in 96-well plates (Corning Costar, #3903). The next day, the medium was replaced and the ADAM10 inhibitor GI254023X (Aobious, #3611; 3 µM) or the ADAM10/ADAM17 inhibitor GW280264X (Aobious, #3632; 3 µM) was added. This fixed concentration was recommended as a standard concentration for metalloprotease inhibition avoiding off-target-effects as reported by [13,30,31,32,33,34]. As these inhibitors were solved in DMSO, the same amount of DMSO was used as the solvent control in all experiments. Two hours later, different concentrations of cisplatin (obtained from the Clinic Pharmacy Services, UKSH, Campus Kiel) at a concentration of 1 mM dissolved in 0.9% NaCl or 0.9% NaCl used as solvent control were added. Cells were treated in three replicate wells. After 48 h of incubation, supernatants were stored at −20 °C for subsequent enzyme-linked immunosorbent assay (ELISA) quantification. Next, the Multiplex Assay ApoLive-Glo^TM^ (Promega, #G6411), which combines detection of viable cells and caspase3/7 activity, was performed as described in the manufacturer’s instructions (TM325). Cell viability was measured as relative fluorescence units using the following excitation sources (Ex) and emission filters (Em) (RFU, 400Ex/505Em) and caspase3/7 cleavage as relative luminescence units (RLU) using the Caspase-Glo^®^ 3/7 Assay System. Both assays were quantified with a microplate-reader (Infinite 200, Tecan, Männedorf, Switzerland). To facilitate comparability to 3D assays, viability and caspase are displayed separately and caspase activity was not normalized to the viability. The means of three separate experiments were plotted as a dose–response curve using four parameter logistic regressions in GraphPad Prism (GraphPad Software, Inc, San Diego, CA, USA). The half minimal (50%) inhibitory concentration (IC_50_) (viability) or the half maximal effective concentration (EC_50_) (caspase3/7) value was calculated and analyzed as described in the statistical section.

#### 2.4.2. Drug Treatment and Analysis of Viability, Cell Death, and Caspase Activity in Spheroids

Cells were seeded with the same cell numbers as described above in ULA black-transparent 96-well plates (Corning, #4520), and imaged daily using NYONE^®^ Scientific (SYNENTEC) to monitor and analyze spheroid formation over time. As the spheroid density may vary between different cell lines, three different focal offsets were used to ensure each spheroid could be captured in focus. On day four, the diameter of spheroids before treatment was calculated using the spheroid quantification v 0.9 (1 channel) application in YT^®^-Software. Afterward, treatment was performed. We carefully aspirated 150 µL medium using a multichannel pipette to leave 50 µL remaining volume, including the spheroid in the plate. We added 50 µL fresh medium. Master mixes of inhibitors or solvent control DMSO and, second, master mixes of cisplatin with NaCl were prepared in medium to reach a final volume of 220 µL/well. The ADAM10 inhibitor GI254023X (Aobious, #3611) or the ADAM10/ADAM17 inhibitor GW280264X (Aobious, #3632) was prepared with a final concentration of 3 µM or the same amount of DMSO as the control. After 2 h of incubation, cisplatin concentrations of 0.1–100 µM were added to the wells in triplicates. As a negative control, the equivalent volume of NaCl was used. The next day, CellTox^TM^ Green cytotoxicity (Promega #G8743) stain was added in a final dilution of 1:2000 and imaged 3 h later using NYONE^®^ Scientific and the following settings: brightfield (Ex: BF, Em: Green (530/43 nm)), CellTox^TM^ Green (Ex: Blue (475/28 nm); Em: Green (530/43 nm)). For cytotoxicity quantification, the CellTox^TM^ Green signal within the spheroid was analyzed using the spheroid quantification (1F) (v. 0.9) (2 channel) application in YT-Software^®^ using the result “Average fluorescence intensity CH1 BC [1]”. Following a second CellTox™ Green Cytotoxicity Assay measurement in NYONE^®^ 48 h after treatment, endpoint viability and caspase3/7 measurements were performed using two different luminescent assays. For quantification of viability, the RealTime-Glo™ MT Cell Viability Assay (Promega, #G9712) assay was applied and quantified 1 h after incubation at 37 °C using Infinite 200 (Tecan). Next, a Caspase-Glo^®^ 3/7 Assay System (Promega, #G8093) was used at RT, and measured 1 and 2 h after assay application because, for some spheroids, the dynamic range was higher after 2 h, probably due to penetration issues, whereas for others, 1 h was sufficient. The following luminescence filter and integration times were used: for viability, Filter 1 (BLUE2_NB–5000 ms); for caspase measurement, Filter 2 RED_NB–5000 ms) (Infinite 200, Tecan). The means of three separate experiments were plotted as a dose–response curve using four parameter logistic regressions in GraphPad Prism 9, and IC_50_ or EC_50_ values were calculated (GraphPad Software, Inc.). Supernatants of these experiments were stored at –20 °C for subsequent ELISA experiments.

#### 2.4.3. ELISA-Based Quantification of Substrates in Supernatants of 2D Cell Cultures and 3D Spheroids after Drug Treatment

Cell culture supernatants of 2D or 3D cell cultures 48 h after treatment were collected to investigate substrate release. The concentration of HB-EGF was measured as a surrogate for ADAM17 activity. Therefore, the Human HB-EGF DuoSet ELISA (R&D Systems, #DY259B) was used according to the manufacturer’s instructions in NUNC-IMMUNO plates (Thermo Scientific, #442404, Waltham, MA, USA). Using a microplate reader (Infinite 200, Tecan), the optical density (OD) was measured at 450 nm, and HB-EGF concentrations were calculated using MS Excel (2010). The means of three independent experiments + standard error of the mean (SEM) were determined using GraphPad Prism 9 (GraphPad Software, Inc., San Diego, CA, USA).

### 2.5. Statistical Analysis

Unless indicated otherwise, the means of three replicate wells were calculated for each biological experiment using MS Excel (2019). Three independent experiments were performed, and the Gaussian distribution of IC_50_ or EC_50_ values for each treatment (DMSO, GI, or GW) was tested with the Shapiro–Wilk normality test using GraphPad Prism 9 (GraphPad Software, Inc.). Based on the result, either one-way repeated measurement ANOVA (for parametric data of matched datasets) followed by Tukey’s multiple comparison test, Bonferroni’s multiple comparisons, or Friedman’s test followed by Dunn’s multiple comparison post hoc test (for non-parametric matched datasets) was calculated. *p*-values of <0.05 were considered statistically significant. Due to the limited patient material available, only technical replicates were performed; therefore, no statistical analysis was performed. To quantify the strength and direction of the interaction effect between two drugs (for different readouts of 2D and 3D experiments), the drug reduction index (DRI_50_) at 50% effectiveness was calculated [35]. We define DRI_50_ as the fraction of cisplatin concentration required in combination with, e.g., GW280264X compared with cisplatin with solvent control DMSO alone (to reach 50% effectiveness). This definition of DRI_50_ allows for a classification into synergistic (DRI_50_ < 1 or 100%), additive (DRI_50_ = 1), or antagonistic (DRI_50_ > 1) effects. The DRI_50_ was estimated by the EC_50_ ratio (or IC_50_ ratio) of the two fitted standard four-parameter logistic regression dose–response curves (constrained at common maximum, minimum, and slope parameters) using GraphPad Prism 9, and *p*-values for the null hypothesis of DRI_50_ = 1 were calculated.

## 3. Results

### 3.1. ADAM17 Inhibition Reduces Cell Viability and Enhances Cisplatin-Induced Apoptosis

In this study, we systematically evaluated the half minimal (50%) inhibitory concentration (IC_50_) values in five OvCa cell lines applying a fluorescence-based viability assay 48 h after treatment. As described by the vendor, ATCC, or according to the literature, Igrov-1, A2780, and HEY cells are described as cisplatin-sensitive or intermediate-sensitive, and Skov-3 and Ovcar-8 cells as cisplatin-resistant [36,37,38,39,40,41,42]. These data were validated by our IC_50_ values; accordingly, we defined sensitivity as IC_50_ < 10 µM cisplatin and resistance as IC_50_ ≥ 10 µM cisplatin. Additionally, the evaluation of IC_50_ values revealed greatly reduced cell viability using combinatorial treatment with GW280264X (ADAM10 and ADAM17 selective) and cisplatin compared with cisplatin-only treatment. Constant concentrations of the inhibitors (3 µM) were used as described as the best compromise to reach effective inhibition and reduce off-target effects [13,30,31,32,33,34]. Treatment with GW280264X and cisplatin in comparison with cisplatin alone had statistically significantly lower IC_50_ values in all cell lines (*p* < 0.05) (Figure 1), whereas the inhibition of ADAM10 by GI254023X had only a minor effect (Figure 1a). To quantify the effect and direction of the interaction effect between cisplatin and GW280264X, the DRI_50_ was calculated. The DRI_50_ represents, e.g., the cisplatin concentration required in combination with GW280264X compared with cisplatin monotherapy (to reach 50% effectiveness). Thus, the DRI_50_ allows for a classification into synergistic (DRI_50_ < 1), additive (DRI_50_ = 1), or antagonistic (DRI_50_ > 1) effects.

Compared with cisplatin monotherapy, the required concentration of cisplatin in combination with GW280264X to be equivalently effective was, at best, only 7% of the previous cisplatin-only concentration (range 7–50%, depending on cell line) calculated as DRI_50_ values (Figure 1c), indicating strong synergy. Inhibition of ADAM10 by GI254023X had less pronounced effects on all DRI_50_ values (range 61–89%, depending on cell line). In line with this, comparison between both combinatorial treatments, GW280264X + cisplatin and GI254023X + cisplatin, showed a clearer effect of GW280264X (range 31 to 67%, depending on cell line).

Importantly, cisplatin-resistant cell lines were re-sensitized to cisplatin treatment using GW280264X. The sole application of GW280264X was already sufficient to reduce the cell viability by more than 50% in A2780 (*p* < 0.0001) and SKOV-3 cells (*p* = 0.0019), 30% in HEY (*p* = 0.0426) and 20% in Igrov-1 (*p* = 0.0436) and OVCAR-8 (*p* = 0.0002) cells. In contrast, caspase activity was barely induced by applying GW280264X alone. Here, in particular, the combined treatment with cisplatin and GW280264X led to a strong apoptosis induction (Figure 1b). This effect was most prominent once the cisplatin concentration was just high enough to induce cell death. However, concentrations of cisplatin, which did not induce apoptosis in the DMSO control, were sufficient to increase caspase activity when ADAM17 was inhibited. (A2780 1 µM, 2.3-fold increase; HEY 1 µM, 2.4-fold increase; SKOV-3: 5 µM, 3.6-fold increase; OVCAR-8: 7.5 µM, 2.9-fold increase). Consequently, the half maximal effective concentration (EC_50_) differed significantly between the combinatorial and single treatments. In all five cell lines, GW280264X showed a synergistic effect with cisplatin by strongly increasing caspase activity. To reach 50% of the maximum effect, the concentration of cisplatin in combination with GW280264X was reduced down to 18% of the initial concentration (18–61%, depending on cell line) of the cisplatin-only concentration calculated as DRI_50_ values (Figure 1d). Again, cisplatin and GI254023X showed no relevant synergistic effect (all DRI_50_ between 84 and 99%). Finally, in combination with GW280264X, the cisplatin concentration was reduced to values ranging from 38 to 76% for the same effect as cisplatin and GI254023X. To validate the activation of ADAM17 by cisplatin treatment and prove ADAM17 inhibition by GW280264X, we investigated HB-EGF shedding as a surrogate for ADAM17 activity in OVCAR-8 cells using the same treatment scheme as for the live-dead assays, as shown in Appendix A. GW280264X was capable of almost completely reducing basal shedding (without the addition of cisplatin), and diminished cisplatin-induced shedding by almost 80%, whereas GI254023X did not affect induced shedding activity. Taken together, ADAM17 inhibition strongly reduced cell viability even in the absence of cisplatin, and in all five cell lines, the combination of cisplatin and ADAM17 inhibition strongly induced caspase activation.

### 3.2. Generation of Tumor Spheroids and Multi-Content 3D Readout

#### 3.2.1. Generation of Tumor Spheroids

To develop a 3D spheroid model, we initially characterized five OvCa cell lines (Igrov-1, A2780, HEY, SKOV-3, and OVCAR-8) for their capability to form tumor spheroids in ULA plates (Figure 2, upper panel). Whereas Igrov-1 and A2780 cells formed only loose aggregates, SKOV-3 and OVCAR-8 cells formed tight spheroids. HEY cells formed an intermediate type, forming dense aggregates (Figure 2). Application of live–dead stains revealed that SKOV-3 and OVCAR-8 cells were capable of forming a central necrotic core, as observed by propidium iodide (PI, stains dead cells), calcein-AM (stains living cells), and Hoechst33342 (stains nuclei) staining using automated microscopy (Figure 2). For their spheroidal characteristics, we used the HEY, SKOV-3, and OVCAR-8 cell lines for subsequent 3D experiments. To simplify nomenclature, we use the term spheroid throughout the paper, as no universal definition exists for spheroids and aggregates [17,21].

After the establishment of the tumor spheroid model with OvCa cell lines, we wanted to progress toward clinical application using tumor spheroids derived from primary cells. For this purpose, we isolated cells from the tumor tissue and ascites of three late-stage OvCa patients either at the day of surgery or from ascites of recurrent disease and expanded them in cell culture. All four cell populations were seeded onto ULA plates to investigate spheroid formation. Only UF-168 cells formed rather loose aggregates similar to Igrov-1 and A2780. Although all other spheroids formed by primary cells (UF-164 and UF354 tumor and ascites) were more compact and smaller compared with OvCa cell lines, they showed similar characteristics to the tight spheroids of OvCa cell lines, with zonal organization including core formation (Figure 2, lower panel).

#### 3.2.2. Multi-Content 3D Readout

To explore the efficacy of novel treatment regimens in 3D, we established a straightforward workflow to allow for multi-parametric analysis in just one 96-well plate (Figure 3). This was of particular interest given the restricted patient material. Briefly, after four days of spheroid formation, 3D cultures were treated for 48 h (Figure 3). Three consecutive live–dead assays were tested: CellTox^TM^ Green (a fluorescent over time cytotoxicity assay), an endpoint viability, and an endpoint caspase3/7 activity assay. The endpoint assays are both based on luminescent readout but using different luciferases (Figure 3). Moreover, the formation process and treatment effects were imaged daily using automated microscopy, thus enabling the investigation of aggregate sizes and morphology over time. The most accurate and most robust assay combination evaluated in cell lines was subsequently applied to primary cells.

### 3.3. In 3D Cultures, GW280264X Plus Cisplatin Led to a Higher Cisplatin Activity in HEY and OVCAR-8 Cells Compared with Cisplatin Alone

Application of the 3D viability assay in comparison with the 2D model revealed different responses toward cisplatin, mainly in SKOV-3 cells; whereas the IC_50_ in 2D was ~10 µM, that in tumor spheroids was hardly reached applying cisplatin concentrations of 100 µM (Figure 4a). Inhibition of ADAM17 reduced cell viability even in the absence of cisplatin in all investigated spheroids by 15 to 30% compared with the solvent control. In HEY cells, the inhibition of ADAM10 had a minor effect on viability. Interestingly, the general effects of GW280264X alone and in combination were much lower compared with 2D cultures. Even though the IC_50_ curves revealed reduced cell viability in all three cell lines (dashed line), only the IC_50_ values of OVCAR-8 cells were computable and showed significantly reduced IC_50_ values of GW280264X and cisplatin (8.6 µM) vs. DMSO and cisplatin (10.8 µM) or vs. GI254023X and cisplatin (10.0 µM) (Figure 4a). In general, IC_50_ calculations in 3D were not as reliable as in 2D cultures due to curve fitting and the higher resistance potential: no plateau was reached, and supra-physiological concentrations were thought not to provide relevant information. As this assay seemed less robust compared with the other two (Caspase-Glo^®^ 3/7 Assay System and CellTox™ Green Cytotoxicity Assay measurement), this viability assay was discontinued for primary cells.

In concordance with the 2D results, the single application of cisplatin in the absence of inhibitors did not significantly increase caspase activity in spheroidal HEY and OVCAR-8 cells, but dramatically increased caspase activity in the presence of GW280264X once cisplatin was added, even at non-effective levels of cisplatin (Figure 4b). Cisplatin-induced ADAM17 activity and selective inhibition by GW280264X were proved by examining HB-EGF shedding as a surrogate for ADAM17 activity (Appendix A). HB-EGF levels in the supernatants of cisplatin-treated OVCAR-8 spheroids were strongly increased. This enhanced shedding was almost completely mitigated using GW280264X but not by GI254023X (*p* < 0.05). In HEY cells, the combinatorial effect of GW280264X and cisplatin was most pronounced using 2.5 µM cisplatin (two-fold increase compared with cisplatin only; *p* < 0.001).

Interestingly, lower concentrations of cisplatin (1 µM), which were insufficient to increase cell death but induced early apoptotic signaling in the absence of inhibitors, strongly triggered apoptosis when combined with GW280264X (*p* < 0.001). The combined effect of cisplatin and GW280264X in OVCAR-8 cells initiated at a concentration of 5 µM cisplatin and peaked with the application of 10 µM, where apoptotic cell death was twice as high compared with a single application of cisplatin (*p* < 0.001). Underlining the effect size, the combination of 10 µM cisplatin and 3 µM GW280264X increased cell death around 6–7 times compared with untreated cells. Thus, the EC_50_ of three biological replicates differed significantly (*p* < 0.05). To reach the same DRI_50_ effect of caspase activation, the cisplatin concentration was significantly reduced by inhibiting ADAM17 using GW280264X, 36 and 67% in HEY and Ovcar-8 cells, respectively, indicating strong synergism. This combinatorial effect was not detected in SKOV-3 spheroids, even though being prominent in 2D monolayers, highlighting the importance of substance retesting in a 3D setting.

In summary, the caspase assay in spheroids confirmed the strong combined effects in HEY and OVCAR-8 cells, but not in SKOV-3 cells. Given the divergent results in the viability and caspase assay, it was even more important to use a third assay for 3D cultures robust enough for a subsequent translation to primary cells.

### 3.4. Automated Imaging Confirmed the Combined Cytotoxic Effects of GW280264X and Cisplatin in OvCa Tumor Cell Line Spheroids

To visualize the treatment effects in 3D cultures and enable kinetic measurement of cytotoxicity, we implemented the CellTox^TM^ Green cytotoxicity assay. Using automated microscopy, increased cytotoxicity using cisplatin over time was observed. Figure 5 shows the strongest combinatorial effect after 48 h of treatment.

The DNA intercalating dye CellTox^TM^ Green stain is positive if the membrane integrity is lost, thus indicating late stages of cell death. Consequently, we found CellTox^TM^ Green positivity to be shifted to higher concentrations or later time-points compared with caspase activation, being an early marker of apoptosis (Figure 5). In accordance with caspase activation in 2D and 3D cultures, the inhibition of ADAM17 by GW280264X strongly increased the cytotoxic potential (CellTox^TM^ Green positivity) of cisplatin once its concentration was sufficient to induce cell death. Representative brightfield and CellTox^TM^ Green images, showing strong combinatorial effects of GW280264X and cisplatin, are displayed in Figure 5a,b.

The strongest EC_50_ reduction was detected in HEY aggregates. Here, the inhibition of ADAM17 during cisplatin treatment reduced EC_50_ values by 3.5-fold compared with cisplatin-only treatment. Only 27% of cisplatin was required in combination with GW280264X to produce the same cytotoxic effect compared with cisplatin monotherapy (Figure 5d). Moreover, the inhibition of ADAM17 increased cell death even when cisplatin was absent. OVCAR-8 spheroids generally showed a weaker response compared with HEY aggregates, which is in line with being less sensitive in 2D. Still, EC_50_ was reduced 1.5 times when comparing GW280264X with cisplatin treatment to cisplatin alone (*p* < 0.05). Importantly, the DRI_50_ values of the combined GW280264X and cisplatin treatment were significantly lower compared with cisplatin-only treatment or the combination of cisplatin and GI (*p* < 0.0001), emphasizing a strong synergistic effect in HEY and Ovcar-8 spheroids.

As demonstrated by viability and caspase activity data, SKOV-3 cells demonstrated the strongest resistance potential toward cisplatin among the three cell lines in 3D cultures. Even 10 µM cisplatin did not affect cytotoxicity. Only the very high concentration of 40 µM, which initiated cytotoxicity, induced a combinatorial effect (Figure 5a; bottom right). Interestingly, in this cell line, the sole effect of ADAM17 inhibition strongly increased cell death by around 40% compared with cisplatin-only treatment (*p* < 0.001; Figure 5c).

Taken together, in HEY and SKOV-3 spheroids, the inhibition of ADAM17 increased cell death on its own, but was even more effective in combination with cisplatin. Moreover, OVCAR-8 spheroids displayed strongly enhanced cytotoxicity when combined with cisplatin and ADAM17 inhibition. Generally, the most prominent combined effect was produced at cisplatin concentrations already inducing cell death. Additionally, we found that the CellTox^TM^ Green assay produced robust and reproducible results and was sensitive enough to evaluate and calculate treatment responses, providing an essential prerequisite to implement it in our condensed workflow for primary cells.

### 3.5. Translational Application of Multiplex Workflow Confirms Combinatorial Effect of GW280264X and Cisplatin in Primary OvCa Spheroids

To test our hypothesis in primary tumor material, we isolated cells from tumor tissue and/or the accompanying ascites of three patients. The histopathological background and treatment history are displayed in Table 1. The translation of methods to primary cells can be challenging due to the restricted cell numbers and passage cycles. Therefore, we applied our multiplex readout to enable a time and resource-saving, efficient treatment validation in just one 96-well-plate. Based on the assay results tested in OvCa cell lines, we applied our readout to the most robust assays: caspase3/7 activity and CellTox^TM^ Green. For an overview, we displayed four data sets of caspase3/7 results to compare different responses of primary cells derived from a chemo-naïve patient to primary cells derived from the ascites of a patient with recurrent disease. As a proof of concept, we additionally displayed the data of tumor- and ascites-derived cells from the same patient (UF-354). Here, both caspase3/7 and CellTox^TM^ Green assays are displayed (Figure 6b) to validate that the treatment effect was not assay-dependent, showing similar trends.

Inhibition of ADAM17 during cisplatin treatment in primary cells produced a 1.5-fold increase in caspase3/7 activation using 10 µM cisplatin in tumor cells (UF-354 tumor) and a two-fold increase using 20 µM cisplatin in ascites-derived cells compared with cisplatin monotreatment (UF-354 ascites; Figure 6a). Moreover, patient cells with a higher chemotherapy resistance potential, which did not respond to 10 µM cisplatin (UF-164 and UF-168), were sensitized to cisplatin treatment by application of GW280264X. Importantly, we isolated cells from recurrent ovarian cancer treated with carboplatin and paclitaxel in the first-line setting (UF-164). These primary cells were the most resistant but could be re-sensitized by GW280264X application (Figure 6a).

Using CellTox^TM^ Green staining confirmed the superior effect of cisplatin treatment in combination with GW280264X compared with single cisplatin application in UF-354 cells (Figure 6b). The strongest cytotoxic effect was detected in ascites-derived cells after combined treatment with 20 µM cisplatin and GW280264X compared with cisplatin monotreatment. Representative images of spheroids treated with 20 µM cisplatin are displayed in Figure 6. Combined treatment (Figure 6b, left: lower panel, right image) produced the strongest cytotoxicity compared with monotreatment of cisplatin or GI254023X.

## 4. Discussion

Overall OvCa survival rates are still very low, mainly due to late diagnosis and acquired resistance toward platinum-based therapy, occurring in more than 65% of all patients [43]. The latest developments in OvCa treatment include VEGF inhibitors and PARP-inhibitors as maintenance first-line therapies [44,45]. Resistance on the cellular level is caused by a multitude of mechanisms, e.g., enhanced platinum export via efflux pumps, differential expression of pro- or anti-tumoral proteins, and enhanced activation of growth factor receptors [6,7,8]. Thus, there is an urgent need to develop more effective therapeutic strategies to overcome these challenges. Since several novel drugs have shown promising effects in 2D cultures with established cell lines but have failed to be translated into the clinical setting, we established 3D spheroids and used patient-derived cells from tumor tissue and ascites to explore the efficacy of our combination treatment of cisplatin and ADAM17 inhibition.

It was shown that ADAM17 is a potential sheddase of over 80 substrates [46] of which at least six are capable of binding growth factor receptors such as EGFR. Subsequent phosphorylation of their downstream mediators including extracellular-signal-regulated kinases (ERK), phosphatidylinositol 3-kinase (PI3K)/AKT, signal transducers, and activators of transcription 3 (STAT3) or c-Jun N-terminal kinases (JNK) [12] induced cell proliferation, cell survival, or anti-apoptotic signaling [47,48,49]. Importantly, ADAM17 substrates can be increasingly shed upon cisplatin treatment and lead to the activation of downstream signals of cell survival [11,50]. Due to this interesting interaction, ADAM17-directed antibodies or inhibitory strategies, such as pro-domain inhibition, are being explored [15,51].

Based on our previous publication, the key hypothesis in this study was that ADAM17 can be activated by cisplatin, leading to a decreased responsiveness to cisplatin-induced apoptosis. Accordingly, ADAM17 inhibition sensitizes cells to cisplatin in 3D cell culture systems [11]. Even though the role of ADAM17 in cancer is generally well-studied [46,47,52,53], its impact on resistance mechanisms is not completely understood; especially, the impact of 3D culture condition on ADAM17 and its effects on apoptosis has not yet been characterized [54].

In this study, we found that the induction of apoptosis by cytostatics differs significantly between 2D and 3D conditions. Moreover, 3D culture conditions produced a remarkable impact on the responsiveness to cisplatin-only treatment in the established cell lines and primary cells investigated. These findings might be, at least in part, due to changes in molecular marker profiles [24]. The effect of sole ADAM17 inhibition was also strongly modulated in 3D cultures, rendering the cells less sensitive to GW280264X in 3D compared with 2D; this effect has not been reported elsewhere. However, further validation is required. However, the difference is not an issue of drug penetration into spheroids, as shown by the reduction in ADAM17 activity by its inhibitor. Importantly, the key finding that ADAM17 inhibition sensitizes OvCa cells to cisplatin treatment was validated using our 3D culture system in established cell lines and in primary cancer cells, emphasizing the translational aspect of our work. Interestingly, the cell lines behave differently: OVCAR-8 and HEY cells revealed strong combinatorial effects using GW280264X and cisplatin, whereas SKOV-3 showed minor effects. The strongest combined effect using GW280264X and cisplatin was found after cisplatin initiated its cytotoxic potential. We hypothesize that this effect is due to ADAM17 activation upon cisplatin apoptosis-induction. Cisplatin initiates apoptosis [13,14], followed by loss of membrane integrity and phosphatidylserine (PS) becoming detectable [14] at the outer leaflet of the double lipid layer activating ADAM17 [14]. In cells undergoing apoptosis, ADAM17 activation is triggered in sensitive cells (HEY: 1–5 µM) at lower cisplatin concentrations compared with the more resistant cell line (OVCAR-8 5–15 µM). Moreover, this finding was also evident in the 3D setting, particularly in SKOV-3 cells, which are known to become even more resistant in 3D cultures [55]. Therefore, these spheroids only respond to GW280264X at very high (supra-physiological) cisplatin concentrations, as an initial trigger of ADAM17 activation by apoptosis is required to provoke an enhanced combinatorial effect using GW280264X. However, primary tumor cells show a different intrinsic resistance phenotype compared with established cell lines. As shown in a comprehensive study by Nanki et al., resistance phenotypes vary between histological subtypes and BRCA mutational status [56]. Whereas BRCA1 mutation variant (p.L63*) is rather sensitive to platinum treatment, clear cell carcinomas are highly resistant to chemotherapy. All our patient samples showed wild-type BRCA. Thus, it would be interesting to include primary cells of different histopathological properties, such as clear cell carcinoma in future studies. Furthermore, tumor heterogeneity is an important aspect in solid tumors and especially in ovarian cancers.

Strikingly, in our study, the responses to cisplatin differed between cells derived from tumor and ascites of the same patient, raising the question as to whether this finding constitutes the first steps of resistance evolution during tumor development. Some research evidence shows phenotypic differences between tumor cells and their accompanying ascites tumor cells [57,58]. It is well-known that ADAM17 promotes tumor progression, metastasis, and cell invasion [59,60,61]. As shown for a variety of tumor entities, ADAM17 expression is upregulated in metastatic tumor cells, e.g., McGowan et al. found high ADAM17 expression in lymph node metastases compared with primary breast cancer, supporting the hypothesis that ADAM17 is involved in breast cancer progression [62]. Moreover, ADAM17 expression in metastatic gastric cancer and hepatocellular carcinoma was upregulated [59,63]. As ascites represent the major metastatic route of OvCa progression, we speculate that this mechanism is also true for ascites-derived cells from OvCa patients. Buchanan et al. demonstrated differential ADAM17 levels in primary tumor cells and ascites-derived cells of ovarian cancer patients. Increased shedding activity of ADAM17 was provoked by treatment of OvCa cells with ascites fluid [64]. A variety of ADAM17 substrates, including cell adhesion molecules, such activated leukocyte cell adhesion molecule (ALCAM) and nectin-4, as well as EGFR-ligands such as HB-EGF and AREG, were identified in ascites fluid, highlighting the presence of ADAM17 activity [50,64,65]. Importantly, ADAM17 activity was correlated to an invasive phenotype in OvCa cells due to an increased shedding of the adhesion molecule ALCAM, leading to reduced adhesive properties of these cells [65]. Thus, a higher ADAM17 expression or activity in ascites-derived cells (UF354-ascites) compared with tumor derived cells (UF-354-tumor) could explain the stronger responsiveness of these cells to combination therapy using ADAM17 inhibitor GW280264X and cisplatin compared with cisplatin-only treatment.

Ascites-derived cells from a recurrent ovarian cancer patient after first-line therapy with carboplatin and paclitaxel were investigated. These primary cells revealed a stronger resistance potential compared with primary cells of chemotherapy-naïve patients. Being of particular interest for this study, these cells were re-sensitized to cisplatin using GW280264X in combination with high amounts (20 µM) of cisplatin. Differential regulations of E-cadherin expression between tumor- and ascites-derived tumor cells might lead to a more resistant phenotype [57]. Cheng et al. showed that AREG, the most abundant EGFR ligand in ovarian cancer, which is shed by ADAM17, stimulates ovarian cancer cell invasion by downregulating E-cadherin expression [66]. They showed that treatment with AREG upregulates sprouty RTK signaling antagonist 2 (SPRY2) expression by activating the EGFR-mediated ERK1/2 signaling pathway in ovarian cancer [66]. Taken together, functional inhibition of ADAM17 is most effective once cisplatin has triggered apoptotic signaling. We propose that ADAM17 blocking leads to less ligand shedding, less RTK activation, and thus intensified apoptotic signaling by blocking anti-apoptotic pathways.

Even though the blocking of ADAM17 and its multitude of substrates and downstream effectors might lead to broader effects, several studies revealed that the inhibition of single downstream pathways leads to the concomitant activation of compensator pathways of other RTKs and is not sufficient to reduce tumor growth [46]. Novel studies consider blocking several pathways, for example, the combination of the tyrosine kinase inhibitor sunitinib with phosphoinositide 3-kinase/protein kinase B/Akt/mechanistic Target of Rapamycin (PI3K/AKT/mTOR) inhibitor, everolimus, and proto-oncogene tyrosine-protein kinase SRC inhibitor, dasatinib, showed synergistic antitumor activity in an ovarian cancer model [67]. ADAM17 works upstream of all these pathways and thus might be a valuable alternative target to reduce tumor growth and overcome drug resistance. The transfer of our investigations into 3D culture systems using primary tumors cells enabled us to come very close to the clinical situation, expanding the value of our work.

## 5. Conclusions

The inhibition of ADAM17 (GW280264X) in combination with cisplatin results in the synergistic inhibition of viability, and synergistic enhancement of apoptosis even occurs in primary tumor- and ascites-derived OvCa spheroids, thus being a promising target for future combinatorial treatments.

## Figures and Tables

**Figure 1 cancers-13-02039-f001:**
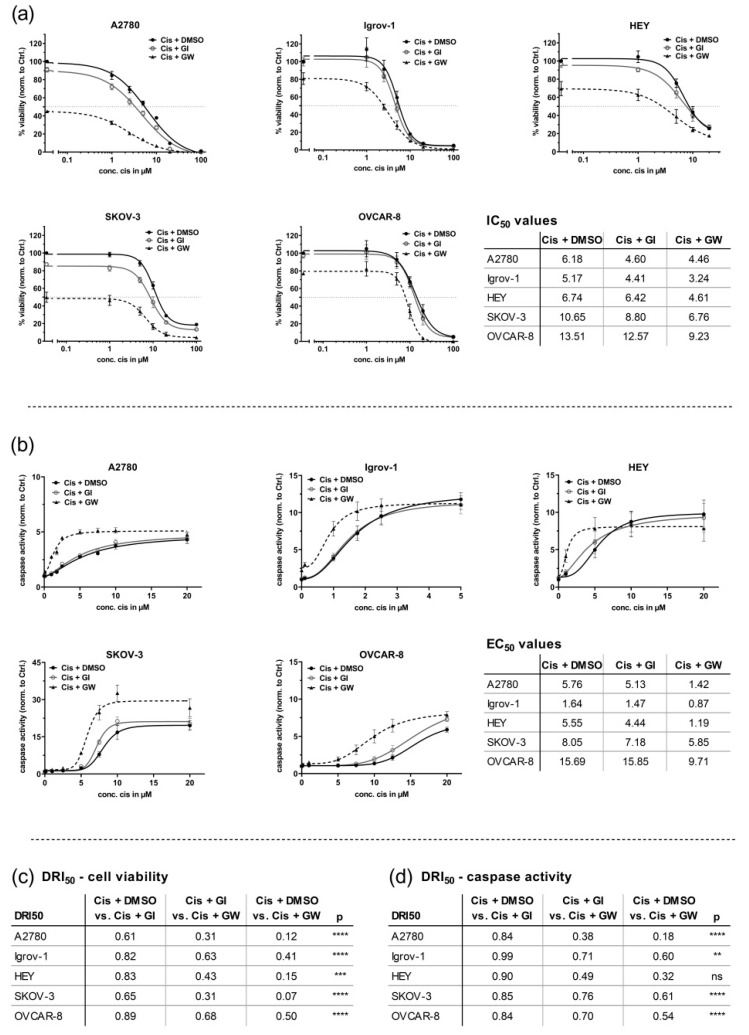
A disintegrin and metalloprotease 17 (ADAM17) inhibition reduces cell viability and enhances cisplatin-induced apoptosis in 2D monolayers. Ovarian cancer (OvCa) cells (A2780, Igrov-1, HEY, SKOV-3, and OVCAR-8) were seeded as 2D monolayers and treated with the indicated concentrations of cisplatin (cis) or NaCl solvent control and 3 µM of either GI254023X (ADAM10 selective), GW280264X (ADAM17 and ADAM10 selective) or the solvent control DMSO. Following 48 h of treatment, the ApoLive-Glo™ Multiplex Assay was used to quantify (**a**) cell viability as relative fluorescence units (RFU) and (**b**) caspase3/7 activation as relative luminescence units (RLU). Data were normalized to control (DMSO and NaCl). The mean (± standard error of the mean (SEM)) of at least 3 biological replicates is displayed. Curve fitting was performed using GraphPad Prism. The half minimal (50%) inhibitory concentration (IC_50_) or the half maximal effective concentration (EC_50_) values were calculated from each curve DMSO, GI254023X (GI), and GW280264X (GW), setting the response of each individual inhibitor (0 µM cis) as the upper baseline value for calculation (i.e., SKOV-3: GI254023X, 85%; GW280264X, 48%). Thus, IC_50_ and EC_50_ values represent the combinatorial effects of cisplatin and inhibitors. Based on the Shapiro–Wilk normality test, IC_50_ and EC_50_ values of the biological replicates were analyzed by ANOVA following Tukey´s multiple comparison test (normally distributed), or Friedman’s test followed by Dunn´s multiple comparison test (not normally distributed). Comparison of IC_50_ or EC_50_ values between DMSO and GW; ns: not significant, ** *p* < 0.01, *** *p* < 0.001, **** *p* < 0.0001. Inhibition of GI254023X did not show significant differences compared with the DMSO control. (**a**) ADAM17 inhibition leads to a strong reduction in cell viability even without the application of cisplatin and a significant reduction in IC_50_ compared with the control (DMSO). (**b**) The combined effect of GW280264X and cisplatin in cisplatin-sensitive cells (HEY) was most pronounced with lower concentrations (1–5 µM) of cisplatin compared with intermediate-sensitive SKOV-3 and OVCAR-8 cells (5–20 µM cisplatin). (**c**,**d**) Drug reduction index (DRI_50_) at 50% effectiveness. As an example, DRI_50_ represents the fraction of cisplatin concentration required in combination with GW compared with cisplatin alone (to reach 50% effectiveness). DRI_50_ can be applied to describe the strength and direction of drug interaction into synergistic (DRI_50_ < 1), additive (DRI_50_ = 1), or antagonistic (DRI_50_ > 1) effects.

**Figure 2 cancers-13-02039-f002:**
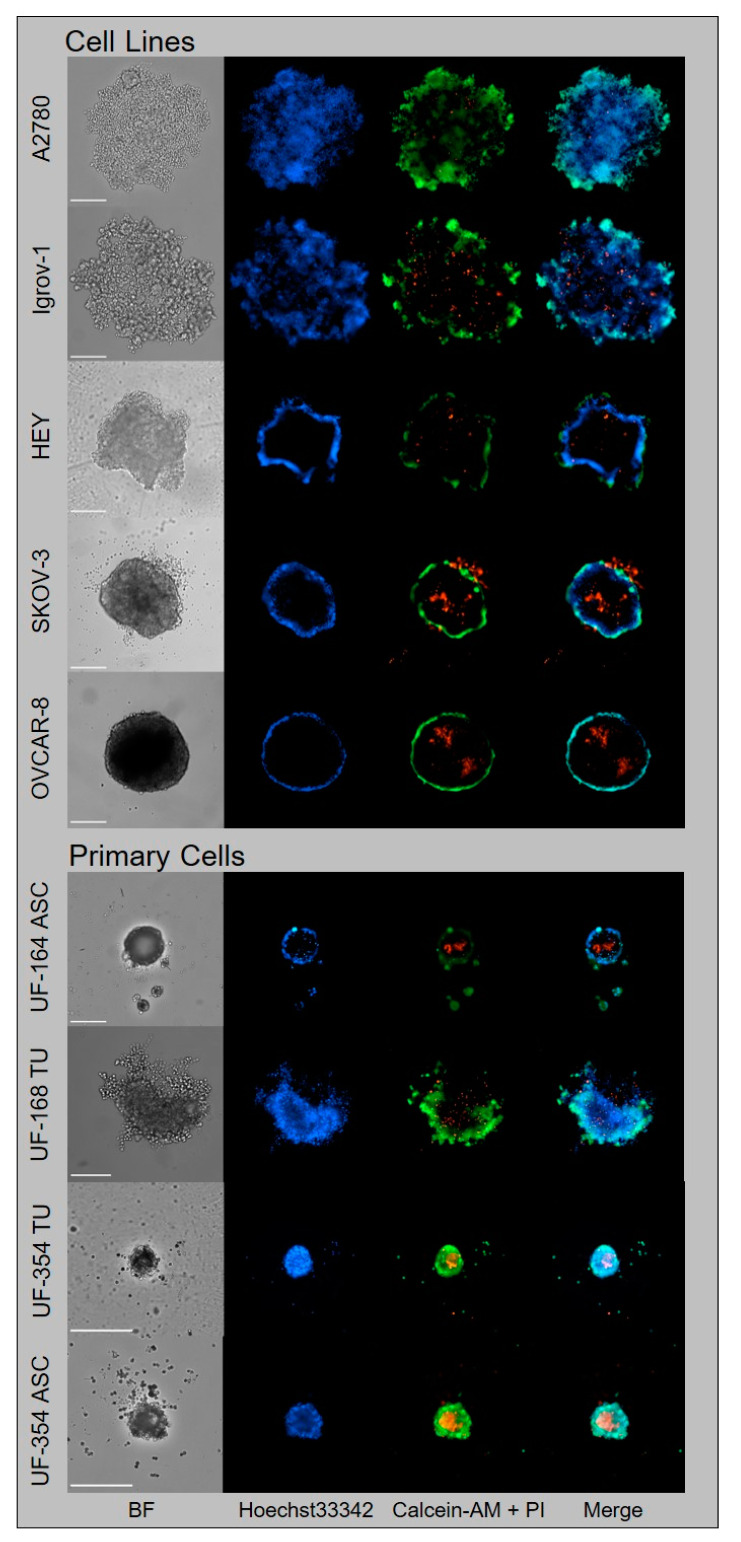
Generation and classification of tumor spheroids using OvCa cell lines and primary OvCa cells. OvCa cell lines and primary OvCa cells from tumor tissue (UF-168 TU and UF-354 TU) or ascites (UF-164 ASC and UF-354 ASC) were seeded in ULA plates and grown for six days for spheroid formation. Triple staining with calcein-AM: living cells, green; propidium iodide (PI): dead cells, red; and Hoechst33342: nuclei, blue revealed different growth types: A2780 and Igrov-1 cells formed loose aggregates indicated by strong central penetration of Hoechst33342 and calcein-AM. HEY cells formed dense aggregates, indicated by the minor central penetration of Hoechst33342 and calcein-AM. SKOV-3 and OVCAR-8 cells formed globe-like dense spheroids with peripheral Hoechst33342 and Calcein-AM staining and a PI-positive core. Spheroids of primary OvCa cells were more compact and showed zonal organization similar to dense spheroids with prominent core formation. Imaging was performed using NYONE^®^ Scientific (SYNENTEC). Scale = 250 µm; magnification 10×.

**Figure 3 cancers-13-02039-f003:**
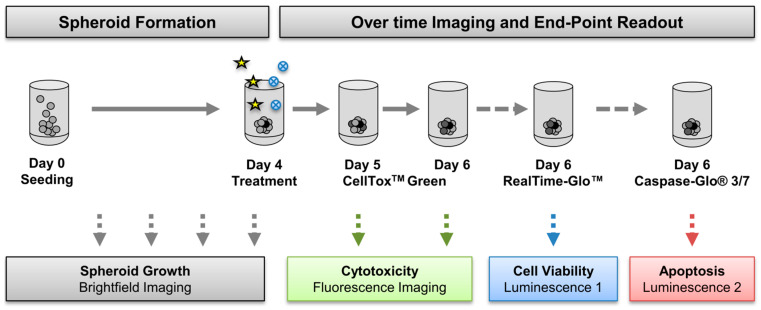
Multi-content 3D readout to test drug combinations. OvCa cells were seeded in ultra-low attachment (ULA) plates and spheroids formed four days before treatment. Daily imaging using NYONE^®^ Scientific (SYNENTEC) recorded the formation process and spheroid morphology and growth over time. On day four, spheroids were treated with cisplatin (asterisks) in absence or presence of metalloprotease inhibitors (blue circles), as described. To allow multiplex readout of three consecutive assays in one 96-well plate, the real-time fluorescence-based cytotoxicity assay CellTox^TM^ Green was first applied on day five and imaged twice: day five = 24 h after treatment, and day six = 48 h after treatment. Next, cell viability and caspase activation were measured using two different luminescent assays to allow for multiplexing using different filters.

**Figure 4 cancers-13-02039-f004:**
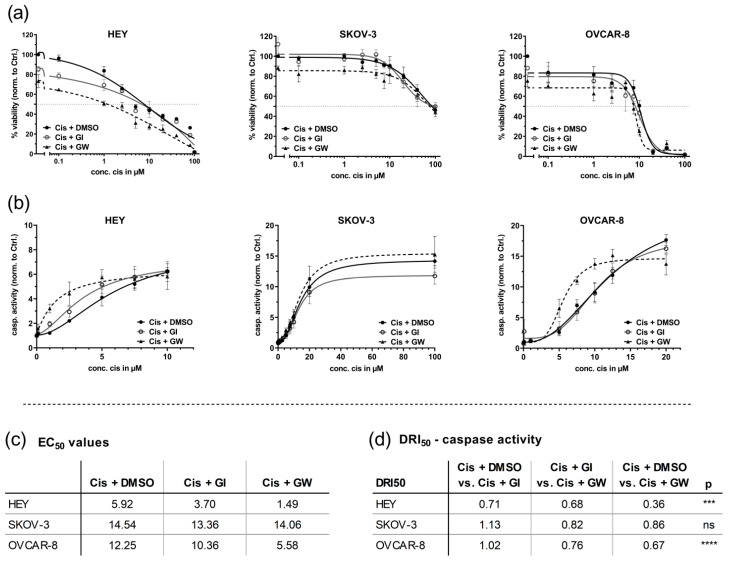
In spheroids of OvCa cell lines, GW280264X plus cisplatin led to a higher cytotoxic effect in HEY and OVCAR-8 cells compared with cisplatin alone. HEY, SKOV-3, or OVCAR-8 cells were seeded in ULA plates with pre-investigated cell densities of 300, 6500, and 4000 cells, respectively, to reach similar spheroid diameters on the day of treatment. On day four after seeding, cells were treated with the indicated concentrations of cisplatin (cis) or NaCl solvent control. For ADAM17 inhibition, GW280264X was used. GI254023X was applied to inhibit ADAM10 and DMSO was used as the solvent control. Curve fitting was performed using GraphPad Prism and EC_50_ values were calculated from each curve of DMSO, GI254023X (GI), and GW280264X (GW), setting the response of each individual curve (0 µM cis) to the lower (EC_50_) baseline value for calculation. (**a**) Quantification of viable cells by the RealTime-Glo™ Assay 48 h after treatment (means of technical replicates ±SD are displayed). (**b**) Apoptosis was measured using the Caspase-Glo^®^ 3/7 reagent. Data of at least three biological replicates were normalized to the control (DMSO, NaCl) and are presented as mean ± SEM. Graphs are displayed until the concentration of cisplatin reached the upper plateau. (**c**) EC_50_ values of caspase3/7; (**d**) drug reduction index (DRI_50_) at 50% effectiveness: DRI_50_ represents, e.g., the cisplatin concentration required in combination with GW compared with cisplatin alone (to reach 50% effectiveness): synergistic (DRI_50_ < 1), additive (DRI_50_ = 1), and antagonistic (DRI_50_ > 1) effects. ns: not significant, *** *p* < 0.001, **** *p* < 0.0001.

**Figure 5 cancers-13-02039-f005:**
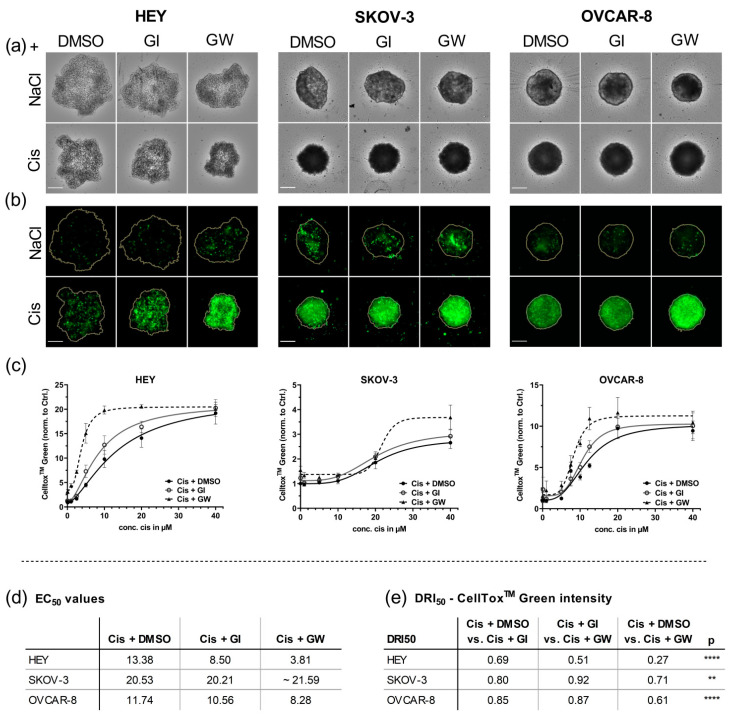
Combined cytotoxic effects of GW280264X and cisplatin in OvCa cell line spheroids can be visualized and quantified using automated imaging. OvCa cell lines were cultured for 4 days in ULA plates for formation of spheroids and treated with the indicated concentrations of cisplatin (Cis) and NaCl. ADAM17 activity was inhibited using GW280264X (GW) and ADAM10 by GI254023X (GI). DMSO was used as a solvent control. At 48 h after treatment, the spheroids were imaged using NYONE^®^ Scientific. (**a**) Representative brightfield (BF) images 48 h after treatment. Displayed are concentrations achieving the strongest combinatorial effects (HEY: 5 µM cisplatin; SKOV-3: 40 µM cisplatin, OVCAR-8: 10 µM cisplatin.). (**b**) Corresponding CellTox^TM^ Green (CTG) images. Orange line indicates spheroid area as determined in the BF image by the spheroid quantification (1F) application of YT^®^-Software. The strongest CTG staining was observed with the combination of cisplatin and GW280264X (lower panel, right image). Objective: 4×, Scale: 250 µm. (**c**) Normalized CTG intensities of at least three biological replicates demonstrate the combined effect of GW280264X and cisplatin. The mean ± SEM is displayed. Curve fitting was performed using GraphPad Prism 9. Based on the Shapiro–Wilk normality test, EC_50_ values of biological replicates were analyzed by ANOVA following Tukey´s multiple comparison test. (**d**) EC_50_ values of CTG; (**e**) drug reduction index (DRI_50_) at 50% effectiveness. Synergistic effect (DRI_50_ < 1), additive effect (DRI_50_ = 1), and antagonistic effect (DRI_50_ > 1). ** *p* < 0.01, **** *p* < 0.0001.

**Figure 6 cancers-13-02039-f006:**
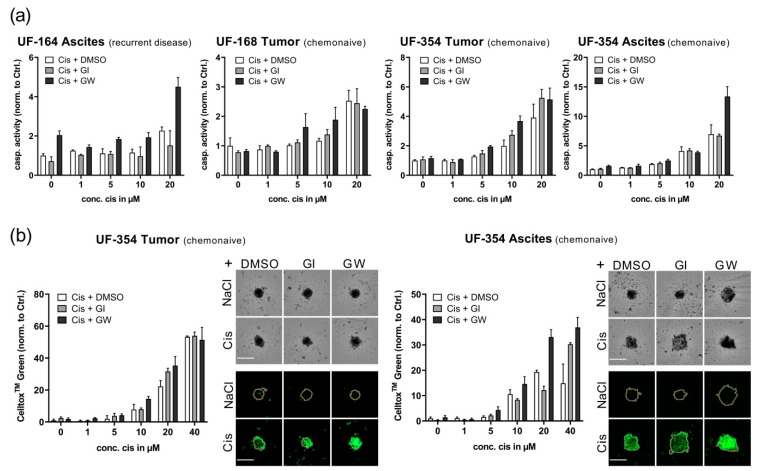
Translational application of multiplex workflow confirms the combinatorial effect of GW280264X and cisplatin in primary OvCa spheroids. Primary cells were isolated from tumor and ascites of three OvCa patients, seeded in ULA plates, and treated after four days of spheroid formation. (**a**) Caspase 3/7 activation 48 h after treatment revealed combinatorial effects by cisplatin treatment (Cis) and inhibition of ADAM17 with GW280264X (GW) compared with sole inhibition of ADAM10 by GI254023X (GI) in tumor-derived cells. DMSO was used as the solvent control. This effect was even more pronounced in ascites. (**b**) CellTox^TM^ Green staining confirmed these combinatorial effects. Representative images show the best combined effect using 20 µM cisplatin (cis). Upper panel: brightfield; Lower panel: CellTox^TM^ Green. Orange line indicates spheroid area. Magnification 4×. Scale: 250 µm. Data are presented as mean + SD of three technical replicates. As for restricted patient material, no statistical analysis was performed using technical replicates.

**Table 1 cancers-13-02039-t001:** Histopathological background and characteristics of patient material used for spheroid generation.

Patients	Subtype	Age at Diagnosis	FIGO	BRCA	Time to Recurrence (Month)
UF-164 ASC	HGSC	69	IIIc	wt	25
UF-168 TU	HGSC	58	IIIc	wt	18
UF-354 TU/ASC	HGSC	56	IIIc	wt	NA

ASC = ascites-derived; TU = tumor-derived; HGSC = high-grade serous carcinoma; FIGO = Fédération Internationale de Gynécologie et d’Obstétrique; tumor classification; BRCA = breast cancer gene; wt = wild type; NA = not applicable. Patient UF-164 was treated with carboplatin and paclitaxel in the first line setting. Primary cells of UF-164 were isolated after tumor recurrence. Tumor cells of patient UF-168 and UF-354 were isolated in a chemonaïve setting.

## Data Availability

The data presented in this study are available on request from the corresponding author.

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
