# Peer review of "ADAM17 Inhibition Increases the Impact of Cisplatin Treatment in Ovarian Cancer Spheroids"

_cancers, 2021, doi:10.3390/cancers13092039_

Round 1
Reviewer 1 Report
In the present manuscript entitled "ADAM17 inhibition increases the impact of cisplatin treatment 2 in ovarian cancer spheroids” by N. Hedermann et. al., authors investigated the efficacy of the combination treatment with the ADAM17 inhibitor, GW280264X, and CDDP in different ovarian cancer cell lines, as well as in three-dimensional tumor spheroids derived from cell lines and from ovarian cancer patients.
The ADAM 17 inhibitor, GW280264X, has been already demonstrated to improve platinum anticancer activity both in vitro, 10.18632/oncotarget.24682, that in vivo, doi: 10.1371/journal.pone.0040597, ovarian cancer models. However, in the present manuscript is tested, for the first time, the efficacy of the co treatment in a patient derived ovarian cancer cells 3D model. This aspect should be emphasized. The histopathological characteristics of the patients should be added in a table. Authors should provide an heterogenous group of derived ovarian cancer 3D model thus to study the behavior of the different models to the co treatment with CDDP and GW280264X. For example, it has been demonstrated, https://doi.org/10.1038/s41598-020-69488-9, that ovarian cancer organoids harboring the BRCA1 pathogenic variant (p.L63*) were more sensitivity to PARP inhibitor, olaparib, as well as to platinum treatment. Conversely, organoids derived from clear cell ovarian cancer were resistant to platinum drugs. Accordingly, the latter could represent an ideal model to revert platinum resistance with the addition of the ADAM 17 inhibitor. In the same way Authors should test this combination in platinum resistant 3D model derived from ovarian cancer cell lines.
It is not clear how the authors choose 3uM as concentration of the adam inhibitor for the co-treatment with CDDP. Moreover, they should please provide combination index graph for all the combination experiments.
Reviewer 2 Report
The paper by Hedeman et al investigates the synergistic effects of ADAM17 inhibitors with cisplatin. For this they use 2D models and 3D models to produce spheroids of ovarian cancer cell lines. The study is novel, well-designed and the data presentation is clear. Chemotherapy resistance is a challenging subject in ovarian cancer and the study has important clinical implications. I only have a few minor comments.
Line 302: “…. day of surgery and expandes them ” should be “expand”
Figure 6: The abbreviations are not explained in this Figure (GW -GW280264X and GI)-
Reviewer 3 Report
The authors aimed to demonstrate three things: combined cytotoxic effect of ADAM17 inhibition and cisplatin on ovarian cancer cell lines; 2D vs. 3D culture models to test drug combinations; and translating their 3D culture approach to primary cells from ovarian cancer patients. The study is well designed and described. While the authors successfully demonstrate the utility of their 3D cell culture model and its translational efficiency for clinical samples, I have several questions/suggestions regarding their drug combination studies.
- How did the authors decide on a dose of 3uM concentration for the GI and GW inhibitors? How much ADAM17/ADAM10 inhibition is affected in all 5 cell lines at this concentration? Also, how much inhibition is affected in spheroids, as penetration will be an issue here.
- Is the effect of ADAM17 inhibition synergistic or additive to cisplatin in their 2D and 3D cell models? The authors should use varying concentrations of the inhibitors in addition to varying concentrations of cisplatin and use specific statistical models that assess synergy (synergy/combination index). This may be useful in the spheroid experiments where varying concentrations of both drugs may have helped achieve toxicity enough to calculate IC50s.
- In assessing the effectiveness of ADAM17 inhibition on cisplatin resistance, the authors should have chosen cell lines that are specifically resistant to cisplatin (OVKATE/FUOV1) or should have established cisplatin-resistant strains for these cell lines. With the current experimental design, it remains unclear if ADAM17 inhibition would re-sensitize cell lines to cisplatin.
Minor comments:
- In various figures, the cisplatin-alone group is mentioned as DMSO instead. This should be changed to cisplatin-alone as referring to them as DMSO suggests that these are untreated cells.
- How many patient samples were used for tumor/ascites spheroid models? What is their treatment history, if any?
- How do the authors explain the lack of effectiveness of ADAM17 inhibition on cisplatin-treated SKOV3 spheroids?
Round 2
Reviewer 1 Report
Although authors have added some interesting data, other aspects need to be further elucidated. In the opinion of this reviewer, the combination index data should be provided in present version of the manuscript.
Author Response
"Please see the attachment."

Reviewer 3 Report
The authors have made several revisions to the text and experimental data based on the comments. I have a few comments.
- Figures 1, 4, 5, and 6: change the group name DMSO to cis + DMSO in the IC50 tables as well.
- Figure 6, were statistical tests performed for this set of data? Please indicate which data points are significantly different. Also, the spheroids derived from UF-354 ascites respond better to the cis + GW drug combination (vs. DMSO and GI) than the spheroids derived from UF-354 tumors. Please discuss this.
- Consider separating the drug treatment portion into a separate section as cells/spheroids used for both the viability/toxicity assays and HB-EGF ELISA were treated the same way. In the present form, treatment conditions for cells and spheroids used in the ELISA are difficult to ascertain.
- Please add the concentration of GW and GI used in the figure legend for S1.
- The manuscript needs language editing. There are instances of punctuation, grammar, and word usage errors. For instance, in line 154, state of the art should be changed to standard/recommended.
Author Response
"Please see the attachment."
